# Double-slit photoelectron interference in strong-field ionization of the neon dimer

Maksim Kunitski[1], Nicolas Eicke[2], Pia Huber[1], Jonas Köhler[1], Stefan Zeller[1], Jörg Voigtsberger[1], Nikolai Schlott[1], Kevin Henrichs[1], Hendrik Sann[1], Florian Trinter[1], Lothar Ph.H. Schmidt[1], Anton Kalinin[3], Markus S. Schöffler[1], Till Jahnke[1], Manfred Lein[2] & Reinhard Dörner [1]

Wave-particle duality is an inherent peculiarity of the quantum world. The double-slit experiment has been frequently used for understanding different aspects of this fundamental concept. The occurrence of interference rests on the lack of which-way information and on the absence of decoherence mechanisms, which could scramble the wave fronts. Here, we report on the observation of two-center interference in the molecular-frame photoelectron momentum distribution upon ionization of the neon dimer by a strong laser field. Post-selection of ions, which are measured in coincidence with electrons, allows choosing the symmetry of the residual ion, leading to observation of both, *gerade* and *ungerade*, types of interference.

[1] Institut für Kernphysik, Goethe-Universität Frankfurt am Main, Max-von-Laue-Straße 1, 60438 Frankfurt am Main, Germany. [2] Institut für Theoretische Physik, Leibniz Universität Hannover, Appelstraße 2, 30167 Hannover, Germany. [3] GSI-Helmholtz Center for Heavy Ion Research, Planckstraße 1, 64291 Darmstadt, Germany. Correspondence and requests for materials should be addressed to M.K. (email: kunitski@atom.uni-frankfurt.de) or to R.D. (email: doerner@atom.uni-frankfurt.de)

**W**ave-like behavior of microscopic particles, e.g. interference, and, in general, the particle-wave duality is "the mystery", as stated by Feynman[1], "which is impossible, absolutely impossible, to explain in any classical way, and which has in it the heart of quantum mechanics". The double-slit experiment, which was originally conducted by Young in the early 1800s to prove the wave nature of light, has been widely utilized to learn about different aspects of this "mystery". In the 1960s it was realized that the double-slit experiment can be performed on the molecular level by exploiting two sites of a diatomic molecule as coherent electron emitters[2]. Liberation of an electron from such a system – for instance, upon single-photon absorption – will result in interference of two partial waves, spatially separated by the molecular bond length. Two-center photoelectron interference upon single photon ionization (weak field regime) has been observed for $H_2$[3–6], $N_2$[5,7–10], $O_2$[11], weakly bound $Ne_2$[12], and even for polyatomic molecules such as simple hydrocarbons[13].

Recent theoretical works[14–16] have suggested that this concept can be transferred to strong-field ionization and thus ultrafast time scales. Reasons possibly contradicting this generalization are that strong-field ionization is often described by tunneling ionization, where the tunnel exit and, thus, the birth position of the electron is not at the atomic centers but at a distance comparable to the extent of the molecule. Furthermore, the electrons are accelerated in the laser field and reach their final momentum only at an even larger distance. Thus, it might not be obvious which momentum is relevant for the spacing of the interference fringes. The presence of the Coulomb field of the parent ion during the acceleration by the laser field causes another potential problem for double-slit interference as it leads, in particular for linearly polarized light, to massive deformation of the phase front of the emitted electron. Experimental support, so far, for double-slit type interference in the strong-field context is suppression of the ionization efficiency of the oxygen molecule[17] and xenon dimer[18] as compared to xenon atom due to destructive two-center interference. In a pioneering experiment, differences in the photoelectron spectra of non-aligned argon dimers and argon atoms were found and attributed to double-slit interference[19]. More recent work, however, contradicted this conclusion[20] and attributed similar differences to post-collision interaction. In general, spatial photoelectron interferences and symmetry of the corresponding molecular orbital are believed to have a significant impact on many applications in strong-field physics such as imaging of electron density and molecular structure as well as high harmonics generation[21–32].

Here we provide an unambiguous experimental proof that the two-center interference survives strong field ionization despite possible obstacles mentioned above. Unlike the earlier experiments[18–20], our approach provides access to the photoelectron momentum distribution in the molecular frame, which clearly shows an interference pattern upon ionization of neon dimer in both circular as well as linear polarized laser fields. The measured photoelectron distributions are well reproduced by the theory based on the two-center interference, leaving no doubts about the origin of the interference pattern. Moreover, by choosing between two dissociation channels of residual $Ne_2^+$ ion, we have been able to switch the symmetry type of this interference.

## Results

**Molecular double-slit**. The most natural observable to search for double-slit interference is the momentum component of the interfering particle parallel to the distance vector between the two slits. Superposition of two spherical waves $\psi_{1,2} = \frac{1}{|\mathbf{r}|} \cdot e^{i\left(\mathbf{k}\cdot\left(\mathbf{r}\pm\frac{\mathbf{R}}{2}\right)+\phi_{1,2}\right)}$ with momentum $\mathbf{k}$ emerging from two centers

separated by $\mathbf{R}$ results in a probability distribution:

$$P \sim \cos^2\left(\mathbf{k}\cdot\frac{\mathbf{R}}{2} + \frac{\Delta\phi}{2}\right) \tag{1}$$

where $\Delta\phi = \phi_1 - \phi_2$ is the initial phase difference between the two waves. For the optical double-slit both waves are in phase ($\Delta\phi = 0$) and one typically expresses interference in terms of the wavelength $\lambda = 2\pi/|\mathbf{k}|$ and the angle $\vartheta$ with respect to the normal to $\mathbf{R}$ as $\cos^2\left(\frac{\pi|\mathbf{R}|\sin\vartheta}{\lambda}\right)$.

On the molecular level, the experimental challenge, thus, is to measure the projection of the electron momentum onto the molecular axis $\mathbf{R}$. As the molecules in a gas-phase sample are randomly oriented, this requires measuring the molecular axis for each ionization event in coincidence with the electron momentum. This is possible for neon dimers, which after single ionization dissociate rapidly (see Supplementary Note 1 and Supplementary Fig. 2) along the bond axis into $Ne^0$ and $Ne^+$; thus detection of the emission direction of the $Ne^+$ ion in coincidence with the electron emission direction allows obtaining the momentum component of the electron along the bond axis.

**Strong field ionization of the neon dimer**. In the experiment $Ne_2$ was ionized by a 40 fs (FWHM in intensity) 780 nm laser field with intensities of $7.3\times10^{14}\,\mathrm{W\,cm^{-2}}$ (Keldysh parameter $\gamma = 0.72$) in case of circular polarization and $1.2\times10^{15}\,\mathrm{W\,cm^{-2}}$ ($\gamma = 0.4$) for linear polarization. The charged products after ionization were detected in coincidence by COLTRIMS[33]. Here we only consider the single-ionization process that leads to the breakup of the dimer into a singly charged and a neutral neon atom.

The kinetic energy release (KER) acquired by the two neon counterparts after dissociation of the dimer shows two distinct structures around 0.15 eV and 1.3 eV (Fig. 1b). The low-energy part corresponds to liberation of an electron from the $2p\sigma_g$ orbital, which results in direct dissociation of the dimer along its $II(1/2)_g$ ionic state (Fig. 1a, blue path). The KER ~1.3 eV is reached by absorption of one additional photon from the laser field after initial evolution along the $I(1/2)_u$ potential energy curve (removal of the electron from the $2p\sigma_u$ orbital). This transition lifts the dimer up to the $II(1/2)_g$ ionic state, but at a shorter internuclear distance, resulting in gain of higher KER[34] (Fig. 1a, red path, indirect dissociation).

**Double-slit interference in circularly polarized light**. For observing double-slit interference, it is decisive that the initial phase difference $\Delta\phi$ is the same for each event. For the optical case, this is trivially true, while for the homonuclear molecular case this is not always granted, since the electrons emerging from molecular orbitals of *gerade* and *ungerade* symmetries have a $\Delta\phi = \pi$. This converts fringes to anti-fringes and washes away the interference pattern. It is the reason why almost all molecular double-slit experiments are performed on $H_2$ with only one symmetry (compare $N_2$[9] and $Ne_2$[12] for exceptions). As shown in Fig. 1, the dissociation process allows us to separate the two cases of emission from a *gerade* and an *ungerade* orbital by selecting events from a certain region of KER. If both KER regions are selected, implying that both ionization channels are mixed, the photoelectron spectrum in the molecular frame shows no interferences (Fig. 1b, inset). The separation is shown in Fig. 2. The electron removed from the *gerade* $2p\sigma_g$ orbital shows an interference pattern with the minimum along the $k_\perp$ axis at $k_\parallel = 0$ (Fig. 2b, direct dissociation), whereas the ionization from the *ungerade* $2p\sigma_u$ orbital results in a maximum at that position

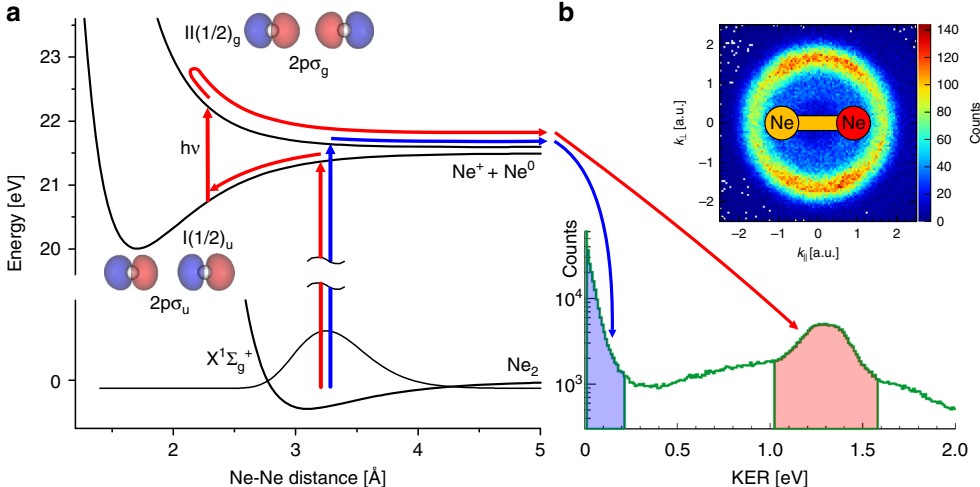

**Fig. 1** Two ionization pathways of neon dimer. **a** Relevant potential energy curves of the neutral and the singly charged neon dimer. Potential energy curves are taken from ref. [43]. Spin-orbit splitting was simulated by separating the asymptotic parts of the I and II curves by 100 meV. **b** Kinetic energy release (KER) for the $Ne^+$-$Ne^0$ dissociation channel. Two dissociation pathways leading to $Ne^+$-$Ne^0$ fragmentation are shown by red (indirect) and blue (direct) arrows. The inset shows the electron momentum distribution in the molecular frame, when both pathways are considered. The red side of the sketched molecule defines the momentum direction of the detected neon ion. $k_\parallel$ and $k_\perp$ are the electron momentum components parallel and perpendicular to the dimer axis, respectively. The corresponding orbitals of the neutral dimer are shown next to the potential energy curves. The ground state energy potential of the neutral dimer $X^1\Sigma_g^+$ as well as the corresponding probability distribution are shown for visualization of the ionization step according to the Franck-Condon principle

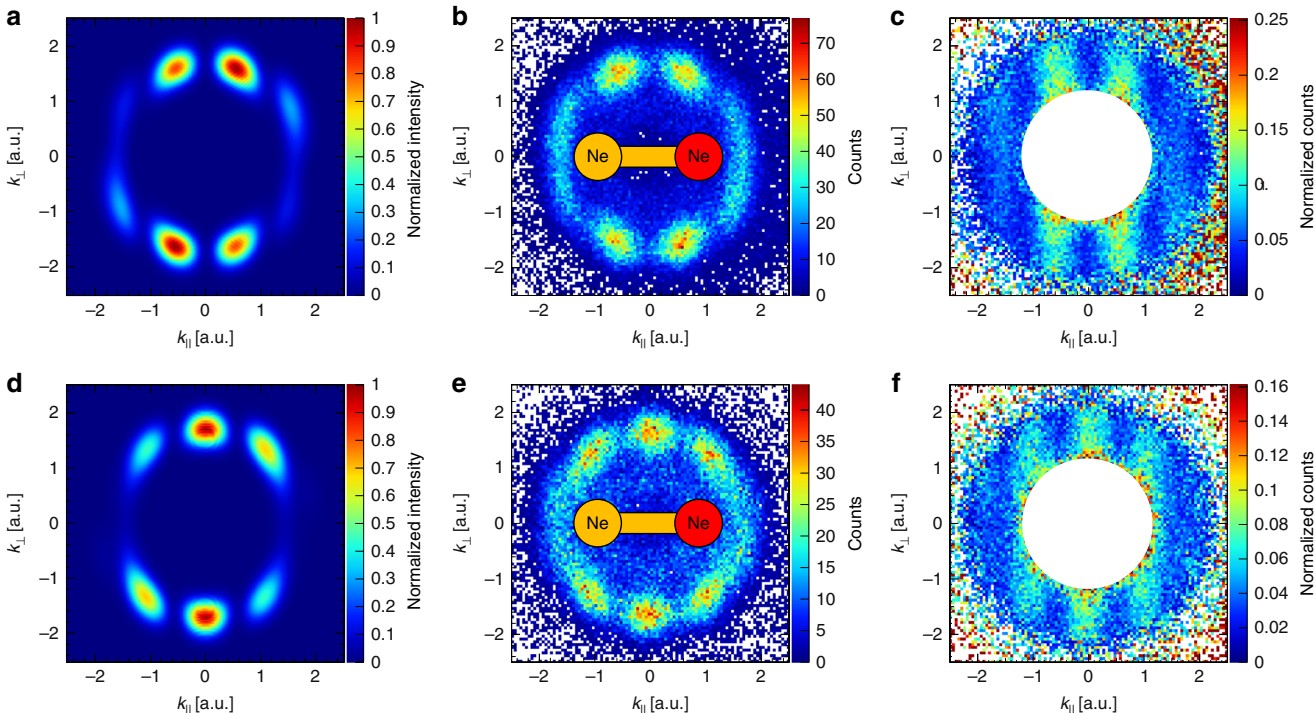

**Fig. 2** Photoelectron momentum distributions of $Ne_2$ in the molecular frame for the circularly polarized light. **a**, **d** are simulated spectra with two coherently superimposed atomic distributions; **b**, **e** – measured for the direct and indirect dissociations, respectively; **c**, **f** are the same as the distributions in **b**, **e** but normalized to the monomer distribution in order to remove ionization weighting of the final momenta. The red side of the sketched molecule defines the momentum direction of the measured neon ion. $k_\parallel$ and $k_\perp$ are the electron momentum components parallel and perpendicular to the dimer axis, respectively

(Fig. 2e, indirect dissociation). This swapped interference behavior is understood given the $p$-character of the orbitals. The interference is indeed seen in the final electron momentum, which is acquired in a laser field after ionization rather than in the initial momentum at the tunnel exit.

Normalization of the electron spectra of the dimer by the single-ionization spectrum of neon atoms, which is recorded in the same measurement, allows us to remove the intrinsic ionization weighting, which has a doughnut-like shape due to tunneling ionization in a circular laser field[35,36]. After such

normalization, the interference stripes perpendicular to the internuclear axis become clearly visible (Fig. 2c, f). These structures resemble the double-slit interferences described by eq. 1 with $\Delta\phi$ being either 0 or $\pi$.

Within our theoretical model, two atomic photoelectron momentum distributions were coherently superimposed in order to reproduce the double-slit interference pattern (Fig. 2a, d). For each atomic spectrum, the time-dependent Schrödinger equation (TDSE) was solved numerically within the single-active-electron approximation. An initial state was chosen to be a $p$ orbital with lobes pointing along the hypothetical internuclear axis, thus, resembling one half of a $2p\sigma$ orbital of the dimer. The final momentum distributions were then projected onto the molecular frame (see Methods). This approach (Fig. 2a, d) accurately reproduces the double-slit interference, apart from the finite contrast in the fringes (see Supplementary Notes 5 and 6, Supplementary Figs. 6-9). This contrast, as the more complete theory indicates, is most likely due to interaction of the outgoing electron with the neighboring atom in the dimer (see Supplementary Note 2 and Supplementary Fig. 3).

**Double-slit interference in linearly polarized light**. Pronounced interference patterns have also been observed upon ionization of the neon dimer in a linearly polarized laser field (Fig. 3). Here the dimer axis was postselected to lie within ±15° to the field polarization direction. The two-center interference changes the photoelectron distribution significantly: for direct ionization of $Ne_2$ the pronounced minimum appears at zero momentum, where naturally, according to the tunneling theory, the maximum of the distribution resides. The experiment with the linearly polarized light shows that the two-center interference also survives among low energy electrons ($k < 0.5$ a.u.) despite their strong Coulomb interaction with the residual ion.

The dependence of the interference on the distance between the slits ($|\mathbf{R}|$) allows, in turn, to utilize the fringe pattern as a tool to measure the bond distance, which is the basis for diffractive imaging. In Fig. 4 we show the sensitivity of this approach. In addition to the interference, the ion momentum gives a second independent measure of the internuclear distance[4]. The II$(1/2)_g$ and I$(1/2)_u$ potentials map the ground-state density distribution of the neon dimer onto the final momentum (energy) of the fragments[37] for the direct and indirect dissociations, respectively. Resolving the distance between the two centers, one can see how

the interference maxima move apart with decreasing separation between the two atomic centers (which corresponds to the higher ion momenta for the direct dissociation and to the lower ion momenta for the indirect one, Fig. 4). This observation is fully reproduced using eq. 1 with either $\Delta\phi = \pi$ or $\Delta\phi = 0$ (Fig. 4b, d). Here, the ion momentum was deduced by mapping internuclear distance to kinetic energy due to motion along the II$(1/2)_g$ and I$(1/2)_u$ potentials. The inverse relation between the internuclear distance and the ion momentum for the indirect dissociation is the proof that the dimer upon ionization initially moves along the binding I$(1/2)_u$ ionic potential energy curve, which has in the Franck-Condon region the opposite slope as the II$(1/2)_g$ potential.

**Internuclear distance distribution of $Ne_2$**. Quantitative estimation of the internuclear distance by applying these two approaches (see Supplementary Note 4 and Supplementary Fig. 5) to the measured distributions in Fig. 4a, c allows to obtain the bond length correlation plots shown in Fig. 5. As one can see, there is a mismatch between potential mapping and interference approaches in determination of the bond length. One possible explanation for this discrepancy is inaccuracy of the potential mapping approach due to the laser induced changes in the II$(1/2)_g$ and I$(1/2)_u$ potentials, similar to the bond-hardening and bond-softening effects in $H_2^+$ [38–40]. Namely, a high laser field might shift down the II$(1/2)_g$ and lift up the I$(1/2)_u$ potential curves resulting in a lower (higher) ion momentum after the direct (indirect) fragmentation with respect to the unperturbed potentials. Since we used the unperturbed potential curves, the deduced bond lengths appeared to be larger for direct dissociation and lower for indirect one than those upon consideration of the real field-dressed potentials.

Assuming that the two-center interference method provide more accurate estimation of the internuclear distances upon ionization, there is still a question why the corresponding length distribution (ca. 2.8–3.25 Å) is lower than the expectation value of the ground state internuclear distance of 3.337 ± 0.001 Å obtained by means of high resolution spectroscopy[41]. The answer to this question depends on the dissociation channel. In case of direct fragmentation the lowering of the II$(1/2)_g$ potential by a laser field results in lower ion momenta upon dissociation. The part of the ground state distribution corresponding to larger internuclear distances, thus, acquires very tiny momenta (<3.5 a.u.) along the

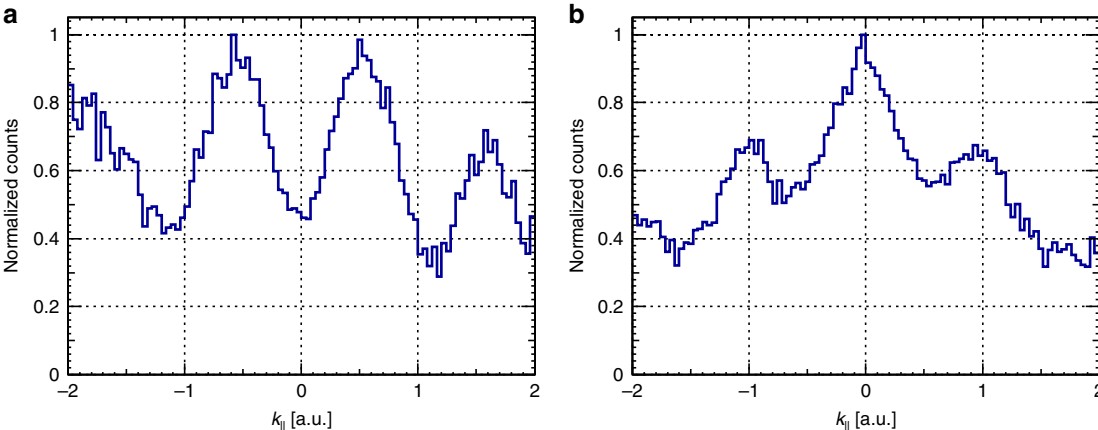

**Fig. 3** Photoelectron momentum distributions of $Ne_2$ for linearly polarized light. Normalized photoelectron momentum distributions projected to the molecular axis for the direct (**a**) and indirect (**b**) dissociation pathways in case of linearly polarized light. The spectra were generated by selecting only the events, where the dimer axis lies within ±15 to the polarization direction of the laser field. In addition, the ion momenta were limited to 4.5–16 a.u. and 39–45 a.u for the direct and indirect dissociation, respectively. Original spectra were divided by the corresponding spectrum of the monomer in order to remove the ionization weighting. $k_{\parallel}$ is the electron momentum component parallel to the dimer axis

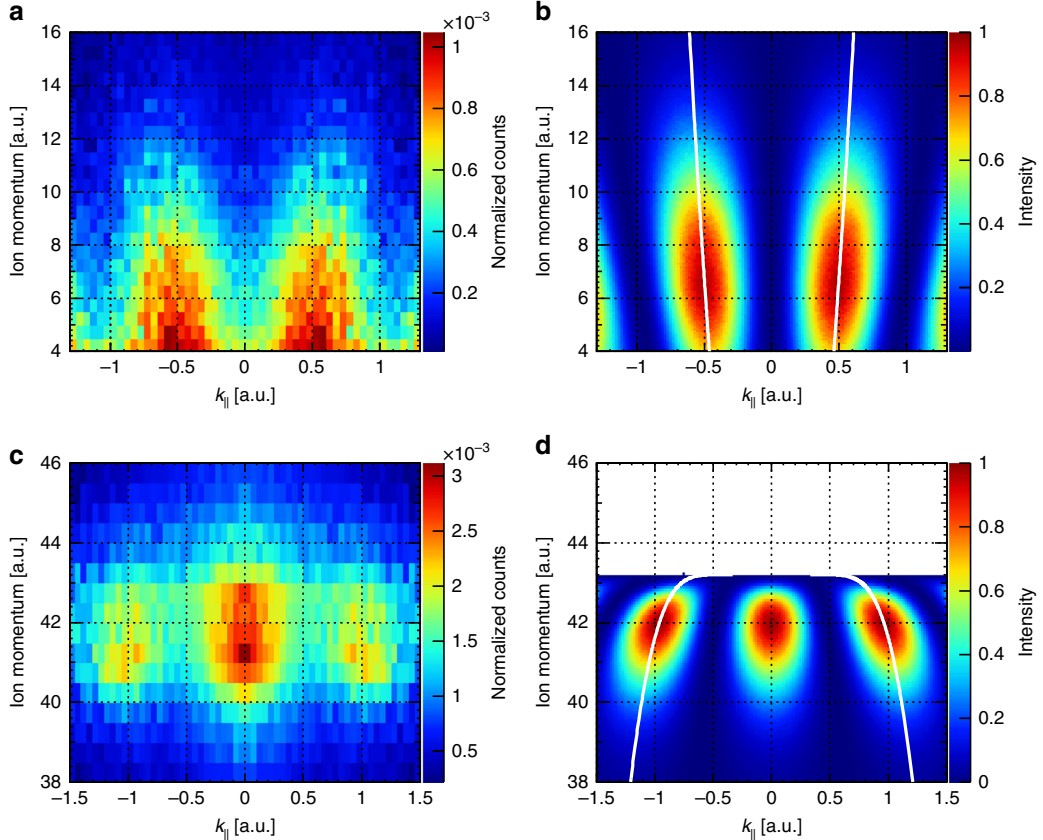

**Fig. 4** Dependence of the two-center electron interference on the internuclear distance. The internuclear distance is encoded in the momentum of the ion. **a**, **c** – experimental distributions for the linearly polarized light (symmetrized with respect to the $k_{\parallel} = 0$ line) for direct and indirect dissociation pathways, respectively; **b**, **d** classical simulations using the $II(1/2)_g$ and $I(1/2)_u$ potential energy curves, respectively (Fig. 1). Each row in **a**, **c** was divided by the spectrum of the monomer in order to remove the ionization weighting. For the eye guidance, the white lines on the simulated distributions show dependence of the fringe position on the ion momentum. The dimer axis was selected to lie within ±15° to the field polarization direction. $k_{\parallel}$ is the electron momentum component parallel to the dimer axis

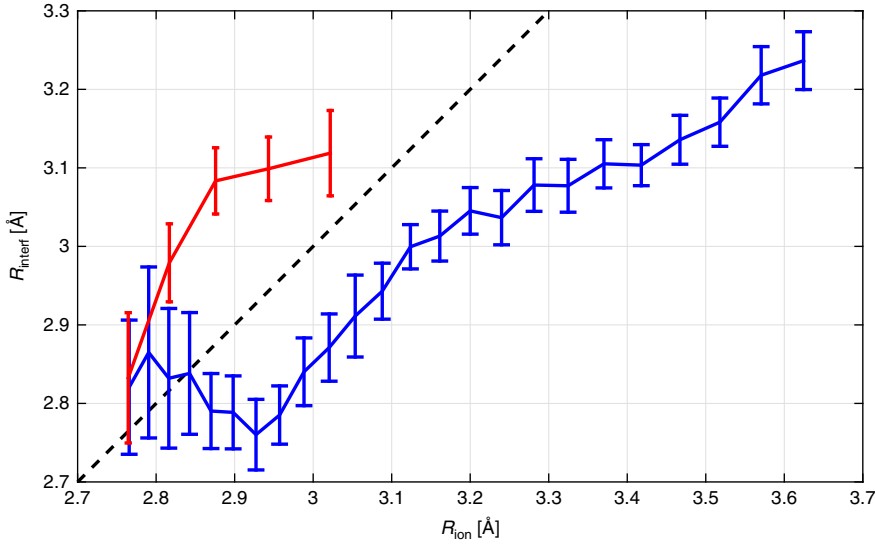

**Fig. 5** Comparison of the two methods in determination of the bond length. $R_{ion}$ is the bond length of $Ne_2$ obtained from the ion momentum after the direct (blue) and indirect (red) fragmentations. $R_{interf}$ is the bond length obtained from the corresponding two-center interference pattern. Statistical error bars for the $R_{interf}$ correspond to the 1σ confidence interval. The black dashed diagonal line visualizes the exact matching dependence

shallow field-dressed dissociation potential and does not fulfill the momentum requirement for being considered as the direct ionization channel (3.5–17 a.u., see Methods). In case of indirect dissociation, the vibrational wave packets started to move along the $I(1/2)_u$ potential at large internuclear distances take longer to reach the Franck-Condon region for one-photon transition to the dissociative $II(1/2)_g$ potential. If the wave packet fails to reach this region within a laser pulse, the $Ne_2^+$ ion remains bound and, thus, undetected in the indirect channel. In both cases, the larger internuclear distances become suppressed.

## Discussion

We have observed two-center photoelectron interference upon ionization of the neon dimer by a strong laser field. Unlike previous double-slit experiments in laser fields, we have measured the two-dimensional photoelectron momentum distribution in the molecular frame. This distribution shows distinct fringe patterns, proving that the two-center interference survives strong field ionization despite some peculiarities of this process. By postselecting the dissociation channel of the dimer we were able to choose the orbital the electron was liberated from. This post-selection allows for switching between *gerade* and *ungerade* interference patterns in the molecular-frame photoelectron distribution, depending on the orbital parity. Turning it around, one can obtain the phase along the molecular orbital by considering the corresponding interference pattern. Another application of the two-center interference is the measurement of the bond length. Moreover, since an ion momentum during dissociation is determined by the corresponding ionic potential, the dependences shown in Fig. 4 might be used for revealing these potential energy curves in the Franck-Condon region. The finite contrast in the interference pattern is most likely related to interaction of the liberated electron with the neighboring neutral atom in the dimer. Extension of the molecular double-slit experiment to strong-field ionization allows for better understanding of different aspects of molecular ionization such as ionization probability and photoelectron angular distribution and opens up ways - ultimately time-resolved - of probing fundamental concepts of quantum mechanics.

## Methods

**Experimental**. The neon dimers were prepared in a molecular beam under supersonic expansion of gaseous neon at a temperature of 60 K through a 5 μm nozzle (see Supplementary Figure 1). The nozzle temperature was stabilized within ±0.1 K by a continuous flow cryogenic cryostat (Model RC110 UHV, Cryo Industries of America, Inc.). The optimum dimer yield was found at a nozzle back pressure of 3 bar. Neon dimers were selected from the molecular beam by means of matter wave diffraction using a transmission grating with a period of 100 nm. The selection allowed increasing the relative yield of $Ne_2$ from typically 2%[12] to 20% with respect to the monomer.

The neon dimers were singly ionized by a strong ultra-short laser field (40 fs -FWHM in intensity -, 780 nm, 8 kHz, Dragon KMLabs). The field intensities were $7.3 \times 10^{14}$ W cm$^{-2}$ (Keldysh parameter $\gamma = 0.72$) in case of circular polarization and $1.2 \times 10^{15}$ W cm$^{-2}$ ($\gamma = 0.4$) in the experiment with linearly polarized light. The 3D-momenta of the ion and the electron after ionization were measured by cold target recoil ion momentum spectroscopy (COLTRIMS). In the COLTRIMS spectrometer a homogeneous electric field of 16 V cm$^{-1}$ for circularly polarized light, or 23 V cm$^{-1}$ in case of linearly polarized laser field, guided the ions onto a time- and position-sensitive micro-channel plate detector with hexagonal delay-line position readout[42] and an active area of 80 mm. In order to achieve $4\pi$ solid angle detection of electrons with momenta up to 2.5 a.u., a magnetic field of 12.5 G was applied within the COLTRIMS spectrometer in the experiment with the circularly polarized laser field. In the case of linearly polarized light a magnetic field of 9 G was utilized. The ion and electron detectors were placed at 450 mm and 250 mm, respectively, away from the ionization region.

**Molecular frame photoelectron spectra**. The photoelectron momentum distributions with respect to the molecular axis shown in Fig. 2 were generated in the following way. Initially, the ions were assigned to the one of the two breakup channels, direct and indirect, by requiring the magnitude of the ion momentum to be within 3.5–17 a.u. and 37–46 a.u., respectively. This gating ensure that the ion

comes from the breakup of the dimer along $II(1/2)_g$ state (Fig. 1a). The ionization of atomic neon as well as dissociation over the other potential curves[43] would result in the ion momentum smaller than 3 a.u. Subsequently, only ionization events have been considered, where ion and electron momentum vectors lie within slices along the polarization plane, defined by the conditions $|p_x| < 0.55$ a.u. for electrons as well as $|p_x| < 3.5$ a.u. and $|p_x| < 12.0$ a.u. for ions from the direct and indirect dissociation channels, respectively (the x-direction is the light propagation direction). These conditions ensure that the angle between a momentum vector and the polarization plane does not exceed 45° in the worst case. For the majority of events this angle is, however, smaller than 30°. Both, electron and ion momentum vectors were projected onto the polarization plane. The projection of the ion momentum defines the $k_{||}$ direction, whereas the two components, $k_{||}$ and $k_\perp$, of the electron projection are plotted in Fig. 2. This type of molecular frame transformation avoids nodes along the dimer axis. It does not conserve the product $\mathbf{k} \cdot \mathbf{R}$, but the loss of contrast in the interference patterns is negligible. Another type of transformation, a natural one, where the ion momentum vector, not its projection, defines the $k_{||}$ direction is presented in the Supplementary Note 3 and Supplementary Fig. 4.

**Theory**. Starting from a $2p$ atomic orbital aligned along the molecular axis, we solve the three-dimensional TDSE in single-active-electron approximation with the split-operator method on a Cartesian grid with 512 points in each dimension, a grid spacing of 0.25 a.u. and a time step of 0.02 a.u. While propagating up to a final time $T = 1500$ a.u., outgoing parts of the wave function are projected onto Volkov states[44]. The potential for a single neon atom is chosen as in ref. [45] but with the singularity removed using a pseudopotential[46] for angular momentum $l = 1$. The clockwise circularly polarized pulse has a 12-cycle $\sin^2$ envelope and a peak field strength of 0.096 a.u. To obtain the momentum distribution for the dimer we multiply two copies of the atomic distribution by $e^{\pm i\mathbf{k}\cdot\mathbf{R}/2}$, respectively ($|\mathbf{R}|/2 = 2.93$ a.u.) and then add them coherently with an additional factor of $\pm 1$ depending on the type of interference, *gerade*, or *ungerade*. To account for different possible orientations of the dimer with respect to the polarization plane, we vary the angle between them in 8 steps to cover a range from 0 to 45°, project the molecular photoelectron momentum distribution (PMD) onto the polarization plane and add these projections together with their geometrical weights. The PMDs are then averaged over the ATI peaks to obtain the final distributions shown in Fig. 2.

## Data availability

The data that support the findings of this study are available from the authors on reasonable request.

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

## Acknowledgements

We acknowledge financial support in the frame of the Priority Programme "Quantum Dynamics in Tailored Intense Fields" (QUTIF) of the German Research Foundation.

## Author contributions

M.K., P.H., J.K., S.Z., J.V., N.S., K.H., H.S., F.T., L.Ph.H.S., A.K., M.S.S., T.J., and R.D. contributed to the experiment. M.K. and R.D. analyzed the data. N.E and M.L. performed the calculations. All authors contributed to the manuscript.

## Additional information

**Competing interests:** The authors declare no competing interests.

