## [Peer Review File · Nature Communications]

Reviewers' comments:

Reviewer #1 (Remarks to the Author):

In this manuscript the authors show that the Ne dimer acts as a double slit when it is ionized by a strong laser field. They illustrate this by presenting photoelectron momentum distributions, referenced to the internuclear axis of the dimer, that show interference patterns characteristic of a double slit. Furthermore, they show how the interference patterns depend on the internuclear distance. The data that is presented is beautiful, the authors have an excellent track record, and there is little doubt that, as an illustration of basic concepts in quantum mechanics, the work will attract a wide audience. There are however some points in the manuscript that should be clarified:

1. In 2007 members of the same group published a paper in Science (Ref. 5 in the present manuscript) in which they demonstrated that H₂ acts as a double slit when ionized. The present manuscript is not therefore the first illustration of this concept. Even though the relevant papers are cited this makes the start of the abstract and the start of the introduction a little misleading in their current form.

2. In 2016 members of the same group published a paper in Physical Review Letters (Ref. 13 in the present manuscript) which showed double slit type interference patterns in photoelectron momentum distributions from the Ne dimer. The present manuscript is not therefore the first application to the Ne dimer. Again the manuscript is slightly misleading in its current form.

3. The "new" feature of the present manuscript is the fact that ionization is caused by a strong laser field. However, it is not made clear why this is significant or why what is shown here is in any way fundamentally different from what has been seen previously using weak-field ionization. It is stated that "Recent theoretical works have suggested that this concept can be transferred to strong-field ionization and thus ultrafast time scales," and later that "Extension of the "molecular" double-slit experiment to strong-field ionization allows for better understanding of molecular ionization," but this is not explained and there don't seem to be any "ultrafast timescales" anywhere in the work that is presented. The general reader is unlikely to know the difference between strong-field and weak-field ionization and I'm not sure that any reader will understand how we might get a "better understanding of molecular ionization". If the work is to be published in Nature Communications this needs to be made clear. The authors should also clearly set out in what ways the present work differs from what has been presented in Refs. 5 and 13. Is the experimental technique fundamentally different from what has been used before, apart from the fact that the ionization source is different?

4. The authors state that "In a pioneering experiment, differences in the photoelectron spectra of non-aligned argon dimers and argon atoms were found and attributed to double-slit interference. More recent work, however, contradicted this conclusion²⁰ and attributed similar differences to post-collision interaction." However, they do not explain how they were able to overcome these objections in their own work. How can they be sure that their own observations are not a consequence of "post collision interactions"?

In its present form the manuscript leaves too many questions unanswered but if the points above can be addressed then it might be publishable in Nature Communications.

Reviewer #2 (Remarks to the Author):

In the manuscript the authors report on COLTRIMS measurement of two-center interference patterns in the photoelectron momentum distribution resulting from strong-field ionization of Neon dimers. The interference pattern becomes visible after post selection according to KER of events resulting from two different ionization - dissociation pathways. The two channels display phase-shifted interference fringes, visible both for circular- and linear-polarized light. Interestingly, the two channels display opposite dependence of the fringe spacing on the ion momentum, indicating opposite slopes for the relevant potential energy curves. The results shown in the manuscript are simple and beautiful, the manuscript is interesting and would be accessible to a wide readership. However, there are a few concerns that I believe need to be addressed before publication.

My main issue with the current manuscript is that the introduction is very shallow, doesn't put the current work in the wider context of the field, and avoids discussing much literature that is relevant to interference in strong field ionization. For example, in a quick literature search I found these two COLTRIMS papers that discuss strong-field ionization two-center interference: Zhang et al, PRA 97 023417 (2018) and Wen Li et al, PNAS 107, 47, 20219-20222 (2010). Additionally, there are several COLTRIMS works that show that the photoelectron momentum distribution is related to the diffraction pattern of the molecular orbital (see e.g. Meckel et al., Science 13, 2008). Looking at the interference of the ionization from two atoms in a dimer is a longer-bond-length limit of that, as the authors themselves note when discussing the phase difference between the two dissociation paths. And third, there are also quite a few High-Harmonics Generation works that discuss interferences from strong-field ionization of molecules and are relevant to the discussion.

Another point I am missing – also related to the broader context of this work – is what is new about the experiment described here that allowed this striking demonstration of a molecular double slit experiment? The authors refer in the introduction to some effects that might hinder the observation of this interference in strong field ionization, but do not later explain why these difficulties do not apply in this case (or how they are overcome).

In addition there are a few other smaller points:

1. How do you distinguish the signal from the dimers and the monomers?
2. In Figure 1:
 - a. What is the $X1\sigma_g^+$ curve, and how is it relevant to the discussion?
 - b. Two KER ranges are attributed to two dissociation pathways. There are a lot of counts also for other KER ranges, what are these attributed to? These do not display any interference?
3. The authors say that the fringe pattern can be used “as a tool to measure the bond distance”, however no number for the bond length is given. Can such a number actually be extracted from these measurements, or does it require a much more accurate knowledge of various experimental parameters? A quantitative estimation would be especially interesting for Fig.4, if not for the actual bond length then at least for the relative change in bond length for different ion momenta.
4. Page 6: “The finite contrast in the interference pattern has been attributed to interaction of the liberated electron with the neighbouring neutral atom in the dimer”. I didn't find the evidence presented in the paper for that convincing enough. The simulations supporting that in the SI actually resemble the experimental results less than the simulations in the main text, and include some intensity features that don't appear in the experiment.

Reviewer #3 (Remarks to the Author):

The authors describe a very interesting and novel (to the best of my knowledge) experiment in which interference effects in the momentum distribution of electrons from strong-field ionization of Ne₂ dimers are observed. Similar interference patterns have been observed previously, however not in the strong-field regime. The data clearly reveals these interference patterns. Post-selection based on the kinetic energy of the electrons further allows the study of specific dissociation channels, which, in the future, may provide additional insight into (ultra-fast) ionization dynamics.

The manuscript is well written and accessible. The abstract and introduction are appropriate.

I feel that there is one aspect of the manuscript that needs further clarification: It is stated that the measured interference patterns show limited visibility. This is not quantified. Qualitatively, it is attributed to the deformation of the wave front of the electron due to the interaction with the neighboring atom. The advanced theory in the supplement is supposed to take this effect into account. However, the interference patterns calculated with the advanced theory show less resemblance with the data than those calculated with the simpler model. On the other hand, a concern raised in the introduction, namely that the tunneling process might lead to an increased virtual source size, and thus to decreased coherence, is not considered at all when describing the loss in visibility.

Other suggested improvements:

- The method hinges on the rapid dissociation of the Ne dimer. A reference describing this process should be added and relevant timescales should be mentioned.
- Additional information (or an appropriate reference) explaining the intrinsic ionization weighting and its doughnut like shape in a circular laser field, would be appreciated.
- Experiments with linear and circular polarization have been performed. It is not well explained whether this yields additional information. While in the introduction it is stated that linearly polarized light would result in a massively deformed wave front of the out-going electron, this is not discussed in the main part of the manuscript (or in fig. 3).
- What was the relative yield of the Ne₂ dimer prior to matter-wave diffraction?
- A sketch of the apparatus would be appreciated.
- The theoretical calculations (fig 2 a and d, S1 b and d) show horizontal and vertical blue lines on my computer. Are these numerical artifacts or file conversion problems?

Response to referees

for the manuscript entitled "**Double-slit photoelectron interference in strong-field ionization of the neon dimer**"

We would like to thank referees for the thorough reading of the manuscript and valuable comments and suggestions. In the following we address all points of criticism and answer the raised questions. The referee's questions given in blue are followed by our replies.

Reviewers' comments:

Reviewer #1 (Remarks to the Author):

In this manuscript the authors show that the Ne dimer acts as a double slit when it is ionized by a strong laser field. They illustrate this by presenting photoelectron momentum distributions, referenced to the internuclear axis of the dimer, that show interference patterns characteristic of a double slit. Furthermore, they show how the interference patterns depend on the internuclear distance. The data that is presented is beautiful, the authors have an excellent track record, and there is little doubt that, as an illustration of basic concepts in quantum mechanics, the work will attract a wide audience. There are however some points in the manuscript that should be clarified:

1. In 2007 members of the same group published a paper in Science (Ref. 5 in the present manuscript) in which they demonstrated that H₂ acts as a double slit when ionized. The present manuscript is not therefore the first illustration of this concept. Even though the relevant papers are cited this makes the start of the abstract and the start of the introduction a little misleading in their current form.

2. In 2016 members of the same group published a paper in Physical Review Letters (Ref. 13 in the present manuscript) which showed double slit type interference patterns in photoelectron momentum distributions from the Ne dimer. The present manuscript is not therefore the first application to the Ne dimer. Again the manuscript is slightly misleading in its current form.

3. The "new" feature of the present manuscript is the fact that ionization is caused by a strong laser field. However, it is not made clear why this is significant or why what is shown here is in any way fundamentally different from what has been seen previously using weak-field ionization. It is stated that "Recent theoretical works have suggested that this concept can be transferred to strong-field ionization and thus ultrafast time scales," and later that "Extension of the "molecular" double-slit experiment to strong-field ionization allows for better understanding of molecular ionization," but this is not explained and there don't seem to be any "ultrafast timescales" anywhere in the work that is presented. The general reader is unlikely to know the difference between strong-field and weak-field ionization and I'm not sure that any reader will understand how we might get a "better understanding of molecular ionization". If the work is to be published in Nature Communications this needs to be made clear. The authors should also clearly set out in what ways the present work differs from what has been presented in Refs. 5 and 13. Is the experimental technique fundamentally different from what has been used before, apart from the fact that the ionization source is different?

Answer #1-3:

The referee is right that the main message of the current manuscript - and this is what makes it different from previous works - is that interferences are also observed upon **strong field ionization** of a molecular double-slit. As stated in the introduction, it is not obvious that such type of interferences survive the tunneling ionization process and subsequent quiver motion in the laser and Coulomb fields. The bulk part of introduction tries to address this point: *“Recent theoretical works¹⁴⁻¹⁶ have suggested that this concept can be transferred to strong-field ionization and thus ultrafast time scales. Reasons possibly contradicting this generalization are that strong-field ionization is often described by tunneling ionization, where the tunnel exit and, thus, the birth position of the electron is not at the atomic centers but at a distance comparable to the extent of the molecule. Furthermore, the electrons are accelerated in the laser field and reach their final momentum only at an even larger distance. Thus, it might not be obvious which momentum is relevant for the spacing of the interference fringes. The presence of the Coulomb field of the parent ion during the acceleration by the laser field causes another potential problem for double-slit interference as it leads, in particular for linearly polarized light, to massive deformation of the phase front of the emitted electron.”*

Another finding of the current work, which has not been reported in the literature so far, is switching between different types of interferences by postselection of the residual ion symmetry.

In addition, the interferences are seen in two dimensions, not only in the angle to the molecular axis, which is usually the case for one-photon ionization experiments.

“Ultrafast timescales” is our vision, since strong ultrashort laser fields open up a way for pump-probe experiments. Thus, one might think about some time-resolved experiments (eventually with which-way information) on molecular double-slits.

Spatial interferences, like double-slit one discussed in the current work, are essential for molecular ionization processes, since they might have a significant impact on outcome of ionization (for instance, ionization probability) and tailor photoelectron angular distributions.

Please see the answer #4 below for the changes made in the manuscript.

4. The authors state that “In a pioneering experiment, differences in the photoelectron spectra of non-aligned argon dimers and argon atoms were found and attributed to double-slit interference. More recent work, however, contradicted this conclusion²⁰ and attributed similar differences to post-collision interaction.” However, they do not explain how they were able to overcome these objections in their own work. How can they be sure that their own observations are not a consequence of “post collision interactions”?

Answer #4:

In our work we measured the photoelectron angular distributions in the **molecular frame**, which allowed to observe interferences in the most natural coordinates. This is different to the experiments in ref. 18, 19 and 20 (note new numbering), where the electron distributions are averaged over all molecular orientations. Our photoelectron distributions are well reproduced even by the simple theory based on the double-slit interference of two spherical waves, which should leave no doubts about the origin of the interference pattern.

In order to address the referees concerns presented in the comments #1-4 we have rewritten introduction and conclusion. Namely, the following sentence has been added: “Two-center

photoelectron interference upon single photon ionization (weak field regime) has been observed for H_2^{3-6} , $\text{N}_2^{7-10,5}$, O_2^{11} , weakly bound Ne_2^{12} and even for polyatomic molecules such as simple hydrocarbons¹³.” The last sentence of the introduction has been changed from “Here we report on an experiment that clearly shows the full interference fringes from two-center interference in strong-field ionization” to “Here we provide an unambiguous experimental proof that the two-center interference survives strong field ionization despite possible obstacles mentioned above. Unlike the earlier experiments¹⁸⁻²⁰, our approach provide access to the photoelectron momentum distribution in the *molecular frame*, which clearly shows an interference pattern upon ionization of neon dimer in both circular as well as linear polarized laser fields. The measured photoelectron distributions are well reproduced by the theory based on the two-center interference, leaving no doubts about the origin of the interference pattern. Moreover, by choosing between two dissociation channels of residual Ne_2^+ ion, we have been able to switch the symmetry type of this interference.”

Now the conclusion reads: “To conclude, we have observed two-center photoelectron interference upon ionization of the neon dimer by a strong laser field. Unlike previous double-slit experiments in laser fields, we have measured the two-dimensional photoelectron momentum distribution in the molecular frame. This distribution shows distinct fringe pattern, proving that the two-center interference survives strong field ionization despite some peculiarities of this process. By postselecting the dissociation channel of the dimer we were able to choose the orbital the electron was liberated from. This postselection allows for switching between gerade and ungerade interference patterns in the molecular-frame photoelectron distribution, depending on the orbital parity. Turning it around, one can obtain the phase along the molecular orbital by considering the corresponding interference pattern. Another application of the two-center interference is the measurement of the bond length. Moreover, since an ion momentum during dissociation is determined by the corresponding ionic potential, the dependences shown in Fig. 4 might be used for revealing these potential energy curves in the Franck-Condon region. The finite contrast in the interference pattern has been attributed to interaction of the liberated electron with the neighbouring neutral atom in the dimer. Extension of the “molecular” double-slit experiment to strong-field ionization allows for better understanding of different aspects of molecular ionization such as ionization probability and photoelectron angular distribution and opens up new ways - ultimately time-resolved - of probing fundamental concepts of quantum mechanics.”

In its present form the manuscript leaves too many questions unanswered but if the points above can be addressed then it might be publishable in Nature Communications.

Reviewer #2 (Remarks to the Author):

In the manuscript the authors report on COLTRIMS measurement of two-center interference patterns in the photoelectron momentum distribution resulting from strong-field ionization of Neon dimers. The interference pattern becomes visible after post selection according to KER of events resulting from two different ionization - dissociation pathways. The two channels display phase-shifted interference fringes, visible both for circular- and linear-polarized light. Interestingly, the two channels display opposite dependence of the fringe spacing on the ion momentum, indicating opposite slopes for the relevant potential energy curves. The results shown in the manuscript are simple and beautiful, the manuscript is interesting and would be accessible to a wide readership. However, there are a few concerns that I believe need to be addressed before publication.

My main issue with the current manuscript is that the introduction is very shallow, doesn't put the current work in the wider context of the field, and avoids discussing much literature that is relevant to interference in strong field ionization. For example, in a quick literature search I found these two COLTRIMS papers that discuss strong-field ionization two-center interference: Zhang et al, PRA 97 023417 (2018) and Wen Li et al, PNAS 107, 47, 20219-20222 (2010). Additionally, there are several COLTRIMS works that show that the photoelectron momentum distribution is related to the diffraction pattern of the molecular orbital (see e.g. Meckel et al., Science 13, 2008). Looking at the interference of the ionization from two atoms in a dimer is a longer-bond-length limit of that, as the authors themselves note when discussing the phase difference between the two dissociation paths. And third, there are also quite a few High-Harmonics Generation works that discuss interferences from strong-field ionization of molecules and are relevant to the discussion.

Answer #1:

Indeed, we were not aware about the recent paper on strong field ionization of xenon dimer (PRA 97 023417 (2018)). This reference and others related to the electron density imaging and high harmonics generation (references #18, #21-32, note new numbering) have been added to the manuscript. One sentence in the introduction has been changed accordingly. Now it reads: "Experimental support, so far, for double-slit type interference in the strong-field context is suppression of the ionization efficiency of the oxygen molecule¹⁷ and xenon dimer¹⁸ as compared to xenon atom due to destructive two-center interference." Furthermore, a sentence has been added to the introduction: "In general, spatial photoelectron interferences and symmetry of the corresponding molecular orbital are believed to have a significant impact on many applications in strong-field physics such as imaging of electron density and molecular structure as well as high harmonic generation²¹⁻³²."

Another point I am missing – also related to the broader context of this work – is what is new about the experiment described here that allowed this striking demonstration of a molecular double slit experiment? The authors refer in the introduction to some effects that might hinder the observation of this interference in strong field ionization, but do not later explain why these difficulties do not apply in this case (or how they are overcome).

Answer #2 (please see also answer #4 to referee #1):

In the introduction, we pointed out that the generalization of the double-slit interference towards the strong-field ionization is not straightforward and trivial, because of the nature of tunneling ionization and subsequent propagation of the electron in the combined Coulomb and laser fields. The interference however survives despite these hindrances.

In our work, we have succeeded in observation of interference pattern because we measured the angular distribution of the electrons in the molecular frame, which is different to the experiments in ref. 18, 19 and 20, where the electron distributions are averaged over all molecular orientations. We have rewritten one part of introduction and conclusion. Please see the answer #4 to referee #1.

In addition there are a few other smaller points:

1. How do you distinguish the signal from the dimers and the monomers?

Answer #3

The magnitude of the ion momentum was selected to be higher than 3.5 a.u. in order to ensure that it comes from the breakup of the dimer along $\text{II}(1/2)_g$ state. The ionization of atomic neon as well as dissociation over the other potential curves (see ref. Phys Rev Lett 107, 273002 (2011)) would result in the smaller ion momentum. The following has been added to the “Methods” section (in the “Molecular frame photoelectron spectra” subsection): “Initially, the ions were assigned to one of the two breakup channels, direct and indirect, by requiring the magnitude of the ion momentum to be within 3.5-17 a.u. and 37-46 a.u., respectively. This gating ensure that the ion comes from the breakup of the dimer along $\text{II}(1/2)_g$ state (Fig. 1a). The ionization of atomic neon as well as dissociation over the other potential curves³⁴ would result in the ion momentum smaller than 3 a.u. Subsequently, only ...”

2. In Figure 1: a. What is the $X^1\Sigma_g^+$ curve, and how is it relevant to the discussion? b. Two KER ranges are attributed to two dissociation pathways. There are a lot of counts also for other KER ranges, what are these attributed to? These do not display any interference?

Answer #4

- a) $X^1\Sigma_g^+$ is the ground state potential of the neutral Ne_2 . We have shown it along with the ground state density distribution in order to visualize the Franck-Condon transition, which is, in turn, helpful for visualization of two dissociation paths. The following sentence has been added to the caption of Fig. 1: “The ground state potential of the neutral dimer $X^1\Sigma_g^+$ as well as the corresponding probability distribution are shown for visualization of the ionization step according to the Franck-Condon principle.”
- b) The momentum distributions of the Ne_2 breakup into $\text{Ne}^+ \langle \text{Ne}^0$ with the corresponding energy distributions are shown below (on the left of Fig. A1). As one can see on the left, apart from the direct, indirect breakup channels and monomer ionization, there is also unwanted background coming from false coincidences. The reason for this is that no selection based on momentum conservation can be applied here, since one particle of the reaction is uncharged (Ne^0) and, thus, is not detected in experiment. Another unwanted source of false coincidences is single ionization of ^{22}Ne , which resides very close to the indirect dissociation channel. The events related to single ionization of ^{22}Ne have been cut by the following momentum conditions [in a.u.]: $-1 < p_x < 1$ && $-4 < p_y < 4$ && $-38 < p_z < -30$ and not used during further analysis (see the right of Fig. A1).

Fig. A1. The momentum distribution and the corresponding energy distribution of Ne^+ : left – as it was measured in the experiment, right – after removing events corresponding to the ^{22}Ne single ionization.

The energy distribution shown in Fig.1 is thought to be for qualitative visualization of breakup channels. We have exchanged it however with the right one, where events of ^{22}Ne single ionization are removed (this condition was applied for all experimental momentum distributions in the manuscript). This energy distribution has slightly lower background.

The photoelectron momentum distribution in the molecular frame corresponding to this background shows no interference (see Fig. A2).

Fig. A2. Photoelectron momentum distribution in the molecular frame of the background (ion momentum magnitude has chosen to be higher than 18 a.u. but lower than 35 a.u.).

As seen from Fig. A1 the background close to the indirect breakup channel accounts for about one third of the useful breakup events in this channel.

We have estimated the background influence on the fringe contrast for the indirect channel. For this, we have plotted the component of the photoelectron momentum that is parallel to the molecular axis ($k_{||}$) for both the indirect breakup channel and the background (see Fig. A3 below). The background was chosen by requiring the ion momentum to be within a window of 18-35 a.u. The indirect channel corresponds to the ion momenta in the range of 37-46 a.u. In addition all ion momenta were restricted by $|p_x| < 12$ a.u. The volume of the indirect breakup channel in the momentum space was thus by about 0.83 times smaller than that of the background channel. Assuming the constant density of the false coincidences in the momentum space, one get the following estimation for the background signal in the indirect breakup channel caused by false coincidences: $175(\text{red line}) * 0.83 \approx 145(\text{green lines})$ counts. This background makes up about a half of the indirect channel background, as seen in the $k_{||}$ -projection on the left of Fig. A3.

The influence of the false coincidences on the **direct** breakup channel is negligible because of its tiny volume in the ion momentum space and large amount of measured events (the ratio of the useful events to false coincidences is very high, which can also be seen in the energy distribution in Fig. A1).

Fig. A3. The photoelectron momentum distribution in the molecular frame and the corresponding k_{\parallel} -projection of the region between two dashed lines (1.4-1.9 a.u.). Left: indirect breakup channel with ion momenta between 37 a.u. and 46 a.u. Right: the background (false coincidences) with ion momenta between 18 a.u. and 35 a.u. In addition, the following cuts have been applied: $|p_x| < 0.55$ a.u. for electrons and $|p_x| < 12.0$ a.u. for ions. The volume of the indirect break up in the momentum space of the ion is by 0.83 times smaller than that of the chosen background region. The green line is the estimated background signal caused by false coincidences.

The part of this discussion has been added to the Supplementary Information.

3. The authors say that the fringe pattern can be used “as a tool to measure the bond distance”, however no number for the bond length is given. Can such a number actually be extracted from these measurements, or does it require a much more accurate knowledge of various experimental parameters? A quantitative estimation would be especially interesting for Fig.4, if not for the actual bond length then at least for the relative change in bond length for different ion momenta.

Answer #5

The accurate measurement of the bond length would require a more advanced theoretical model than the simple one used in Fig. 4, since the ionization weighting discussed in the paper changes the shape of the interference fringes. We could partially reduce this weighting by dividing the experimental momentum distribution by the corresponding spectrum of the monomer, however, it was not possible to completely remove it. Moreover, the ionization process depends on

internuclear distance due to change in the ionization potential. This dependence should be considered for accurate estimation of the bond length distribution.

Despite these arguments, we have estimated the bond length for distributions in Fig. 4a,c. These estimations have been done by fitting the fringe distributions corresponding to different ion momenta to the following function:

$$f(k_{\parallel}) = A \cdot \cos^2 \left(\frac{k_{\parallel} \cdot R_{\text{interf}}}{2} + \frac{\Delta\phi}{2} \right) \cdot e^{-\frac{k_{\parallel}^2}{\sigma}} + B,$$

where R_{interf} is the internuclear distance, $\Delta\phi$ is a phase that is 0 for the indirect channel and π for the direct one, σ is the doubled variance of the residual Gaussian distribution (ionization weighting). The Gaussian distribution was used only for the indirect channel, since the direct one has only two fringes that are symmetric with respect to $k_{\parallel} = 0$. A and B are the amplitude and the background, respectively. This is how the typical fits for the direct and indirect fragmentation look:

Fig. A4. Fitted profiles and the corresponding residuals of the momentum distributions from Fig. 4a,c: left – direct fragmentation channel at an ion momentum of 9.25 a.u. gives the bond length of 3.00 Å; right – indirect fragmentation channel at an ion momentum of 41.25 a.u. results in a bond length of 3.10 Å

The Fig. A4 along with the discussion above have been added to the Supplementary Information.

A completely independent way of determining the bond length is via the measured ion momentum. By inverting the energy potentials $\text{II}(1/2)_g$ and $\text{I}(1/2)_u$ we were able to obtain the internuclear distance at the moment of ionization (labeled as R_{ion}). Comparison of the bond length estimated by these two approaches has been added as Fig. 5 to the manuscript (see Fig. A5 below).

Fig. A5. Comparison of the Ne_2 bond lengths obtained from the ion momentum after the direct (blue) and indirect (red) fragmentations (R_{ion}) as well as from the corresponding two-center interference pattern (R_{interf}). Statistical error bars for the R_{interf} correspond to the 1σ confidence interval. The black dashed diagonal line visualizes the exact matching dependence.

The corresponding discussion has been added to the manuscript between Figs. 4 and 5. This new discussion reads:

“Quantitative estimation of the internuclear distance by applying these two approaches (see Supplementary Note 4) to the measured distributions in Fig. 4a,c allows to obtain the bond length correlation plots shown in Fig. 5. As one can see, there is a mismatch between “potential mapping” and “interference” approaches in determination of the bond length. One possible explanation for this discrepancy is inaccuracy of the “potential mapping” approach due to the laser induced changes in the $\text{II}(1/2)_{\text{g}}$ and $\text{I}(1/2)_{\text{u}}$ potentials, similar to the “bond-hardening” and “bond-softening” effects in H_2^+ ³⁹⁻⁴¹. Namely, a high laser field might shift down the $\text{II}(1/2)_{\text{g}}$ and lift up the $\text{I}(1/2)_{\text{u}}$ potential curves resulting in a lower (higher) ion momentum after the direct (indirect) fragmentation with respect to the unperturbed potentials. Since we used the unperturbed potential curves, the deduced bond lengths appeared to be larger for direct dissociation and lower for indirect one than those upon consideration of the real field-dressed potentials.

Assuming that the two-center interference method provide more accurate estimation of the internuclear distances upon ionization, there is still a question why the corresponding length distribution (ca. 2.8-3.25 Å) is lower than the expectation value of the ground state internuclear distance of 3.337 ± 0.001 Å obtained by means of high resolution spectroscopy ⁴². The answer to this question depends on the dissociation channel. In case of direct fragmentation the lowering of the $\text{II}(1/2)_{\text{g}}$ potential by a laser field results in lower ion momenta upon dissociation. The part of the ground state distribution corresponding to larger internuclear distances, thus, acquires very tiny momenta (< 3.5 a.u.) along the shallow field-dressed dissociation potential and does not fulfill the momentum requirement for being considered as the direct ionization channel (3.5-17 a.u., see Methods). In case of indirect dissociation, the vibrational wave packets started to move along the $\text{I}(1/2)_{\text{u}}$ potential at large internuclear distances take longer to reach the Franck-Condon region for one-photon transition to the dissociative $\text{II}(1/2)_{\text{g}}$ potential. If the wave packet fails to reach this region within a laser pulse, the Ne_2^+ ion remains bound and, thus,

undetected in the indirect channel. In both cases, the larger internuclear distances become suppressed.”

4. Page 6: “The finite contrast in the interference pattern has been attributed to interaction of the liberated electron with the neighbouring neutral atom in the dimer”. I didn’t find the evidence presented in the paper for that convincing enough. The simulations supporting that in the SI actually resemble the experimental results less than the simulations in the main text, and include some intensity features that don’t appear in the experiment.

Answer #6

The simulation based on the atomic distributions (Fig. 2 of the manuscript) shows infinite fringe contrast, which is not the case for the experimental momentum distributions. The more complex theory discussed in the supplementary information can reproduce the fact that the experimental contrast is finite by accounting for interaction (scattering) of the outgoing electron wave packet with the neutral atom in the dimer. Therefore, we have concluded that this interaction is responsible for the finite fringe contrast. Another source that reduces the contrast are false coincidences. However, they account for only one-half of the background in the case of the indirect dissociation, and are rather irrelevant for the direct breakup channel.

It is true that the advance theory produces some additional features in the photoelectron momentum distribution that are not observed in the experiment. The reason for this might be an insufficient accuracy of the model potential for the neutral atom that was used in the TDSE calculations. The model potential for the neutral Ne atom is taken from ref. 3 in Supplementary Information. The quality of the potential in this reference was checked by comparing the computed electron scattering cross-sections based on the proposed potential with the experimental ones. This comparison was done however for electrons with energies higher than 50 eV (a momentum of 1.9 a.u.). Moreover, it was stated that the experimental scattering cross-section is not well reproduced by the simulation based on the model potential in the low energy range. Since electrons upon tunneling have even lower energies ($E < 3.4$ eV or $p < 0.5$ a.u.), as can be seen from the experimental momentum distribution perpendicular to the polarization plane (where the laser field is zero, Fig. A6), one might assume that the accuracy of the model potential in the desired electron energy region is not that high. Another possible source of error relates to the fact that the potential of the neutral atom had to be converted into a pseudopotential to make the calculation feasible.

The idea, however, behind using the advance theory was to give qualitative explanation of the finite contrast in the measured interference pattern.

Fig. A6. The photoelectron momentum distribution perpendicular to the polarization plane. “x” is direction of the laser propagation.

Reviewer #3 (Remarks to the Author):

The authors describe a very interesting and novel (to the best of my knowledge) experiment in which interference effects in the momentum distribution of electrons from strong-field ionization of Ne₂ dimers are observed. Similar interference patterns have been observed previously, however not in the strong-field regime. The data clearly reveals these interference patterns. Post-selection based on the kinetic energy of the electrons further allows the study of specific dissociation channels, which, in the future, may provide additional insight into (ultra-fast) ionization dynamics.

The manuscript is well written and accessible. The abstract and introduction are appropriate.

I feel that there is one aspect of the manuscript that needs further clarification: It is stated that the measured interference patterns show limited visibility. This is not quantified. Qualitatively, it is attributed to the deformation of the wave front of the electron due to the interaction with the neighboring atom. The advanced theory in the supplement is supposed to take this effect into account. However, the interference patterns calculated with the advanced theory show less resemblance with the data than those calculated with the simpler model. On the other hand, a concern raised in the introduction, namely that the tunneling process might lead to an increased virtual source size, and thus to decreased coherence, is not considered at all when describing the loss in visibility.

Answer #1

We have calculated the fringe contrast by making use of the projections as shown in Fig. A3. The contrast was found to be $970/330 \approx 2.9$ and $580/245 \approx 2.4$ for the direct and indirect fragmentation channels, respectively (Fig. A7). Determination of the fringe contrast has been added to the Supplementary Information.

Fig. A7. The k_{\parallel} -projections of the photoelectron momentum distribution in the molecular frame for an electron momentum region of 1.4-1.9 a.u., as shown in Fig. A3.

Concerning the performance of the advanced theory, please see the answer #6 to referee #2. The tunneling process, and thus all effects it may bring about, is implicitly included in our TDSE calculations. However, without considering scattering on the neutral atom and coupling between two states of the parent ion it was not possible to reproduce the degree of the contrast seen in the experiment.

Other suggested improvements:

- The method hinges on the rapid dissociation of the Ne dimer. A reference describing this process should be added and relevant timescales should be mentioned.

Answer #2

By taking the potential curve $\Pi(1/2)_g$ from ref. 34 we have classically estimated times the dimer needs to reach an internuclear distance of 10 Å (taken as a reference) during dissociation starting at different internuclear distances within the ground state probability distribution:

Fig. A8. The time the dimer needs to reach an internuclear distance of 10 Å during direct dissociation along the $\Pi(1/2)_g$ potential curve.

Thus, in the worst case, when dimer starts to dissociate along $\Pi(1/2)_g$ at an internuclear distance of 3.8 Å, it takes 1.5 ps to reach 10 Å. The rotational times, however, are much longer. $T_{\text{rot}} \sim 1/(2B) = \mu R^2$, where $B = 1/(2I)$ is the rotational constant in atomic units, and $I = \mu R^2$ is the moment of inertia. For Ar_2 with $\mu = 20.0$ amu and $R = 3.83$ Å, $T_{\text{rot}} = 290$ ps (see ref. PRA, 83, 061403(R), 2011). Given $\mu = 10.0$ amu and $R = 3.2$ Å for Ne_2 , $T_{\text{rot}} \sim 100$ ps.

According to our estimations, the indirect dissociation happens even faster: an internuclear distance of 10 Å is reached within ~200-300 fs. Qualitative explanation for this is that the initial movement on the $I(1/2)_u$ potential should be accomplished within a laser pulse (40 fs, FWHM in intensity), otherwise the vibrational wave packet will not be lifted up to the dissociative $\Pi(1/2)_g$ potential curve, leaving dimer ion bound. The subsequent movement on the $\Pi(1/2)_g$ potential curve is faster than in case of direct dissociation, since the starting internuclear distance is much shorter. The same dissociation path was used for explanation of the 200 fs-long signal depletion in the pump-probe experiment with Ar_2 (see reference 35 in the manuscript). Since the potential energy curves of Ne_2 are similar to those of Ar_2 , but the reduced mass is only a half of Ar_2 , the dissociation dynamics of Ne_2^+ should be even faster.

The discussion above along with Fig. A8 has been added to the Supplementary Information.

- Additional information (or an appropriate reference) explaining the intrinsic ionization weighting and its doughnut like shape in a circular laser field, would be appreciated.

Answer #3

We have added the references 36 and 37 to the sentence “Normalization of the electron spectra of the dimer by the single-ionization spectrum of neon atoms, which is recorded in the same measurement, allows us to remove the intrinsic ionization weighting, which has a “doughnut”-like shape due to tunneling ionization in a circular laser field³⁶⁻³⁷.”

- Experiments with linear and circular polarization have been performed. It is not well explained whether this yields additional information. While in the introduction it is stated that linearly polarized light would result in a massively deformed wave front of the out-going electron, this is not discussed in the main part of the manuscript (or in fig. 3).

Answer #4

The fact that interferences pattern is seen for the linear light as well proves that the effects mentioned in the introduction as possible obstacles, are not that severe. Moreover, the experiment with linearly polarized light provides electron momentum distributions with the low energy region. The low energy electrons also show two-center interference, though their Coulomb interaction with the parent ion is more pronounced.

The following sentence has been added to the manuscript: “. The experiment with the linearly polarized light shows that the two-center interference also survives among low energy electrons despite their strong Coulomb interaction with the residual ion.”

- What was the relative yield of the Ne_2 dimer prior to matter-wave diffraction?

Answer #5

We haven't measured the yield of the Ne_2 prior to diffraction. Usually, however, it is not higher than 2% (ref. 12). One sentence in the “Experimental” subsection of “Methods” has been changed. Now it reads: “The selection allowed increasing the relative yield of Ne_2 from typically 2%¹² to 20% with respect to the monomer.”

- A sketch of the apparatus would be appreciated.

Answer #6

The sketch of the experimental setup has been added to the supplementary information as Supplementary Figure 1. The other figures have been renumbered accordingly.

- The theoretical calculations (fig 2 a and d, S1 b and d) show horizontal and vertical blue lines on my computer. Are these numerical artifacts or file conversion problems?

Answer #7

Indeed, the lines are also seen in high resolution pictures prior to converting to pdf. We have found out that the lines is a result of the anti-aliasing filter during conversion from a vector graphic to a bitmap image. Switching off the anti-aliasing filter removes the artifacts. The figures have been updated accordingly.

Other changes made to the manuscript:

1. We have found a small mistake in the code that normalizes distributions in Figs. 3 and 4. The mistake was corrected and the distributions in Figs. 3 and 4 were updated accordingly. In addition, we have changed the angle condition for the dimer axis from $\pm 20^\circ$ to $\pm 15^\circ$ and restricted the ion momenta to 4.5-16 a.u. and 39-45 a.u for the direct and indirect dissociations, respectively. This allowed for slight improvement of the contrast in the distributions. The following sentence has been added to the caption of Fig. 3: “In addition, the ion momenta were limited to 4.5-16 a.u. and 39-45 a.u for the direct and indirect dissociation, respectively.”
2. The following sentences have been added to the caption of Fig. 4: “The dimer axis was selected to lie within $\pm 15^\circ$ to the field polarization direction.”
3. In the “Theory” subsection of Methods in the sentence “The counterclockwise circularly polarized pulse has a 12-cycle \sin^2 envelope...” the word “counterclockwise” has been replaced by the “clockwise” after reexamination of the projection procedure.

REVIEWERS' COMMENTS:

Reviewer #1 (Remarks to the Author):

I am content that the issues raised in my original report have been addressed and that the manuscript is now publishable in Nature Communications.

Reviewer #2 (Remarks to the Author):

The authors have satisfactorily answered my and the other referees' questions. I recommend publication in Nat. Comms. with a few additional clarifications:

1. Comparison to theory (page 6) – “This contrast, as the more complete theory reveals, is mainly due to interaction of the outgoing electron with the neighboring atom in the dimer” : Since the agreement with the ‘more complete theory’ is not very good, a less strong statement is required here.

2. Results of linear polarization:

a. “... shows that the two-center interference also survives among low energy electrons” – What is considered low energy? Are the electrons measured all / mostly low energy? (Impossible to know from this data, since the third momentum axis is missing)

b. “... despite their strong Coulomb interaction with the residual ion” - In linear polarization there are two contributions to the signal – ‘direct’ and ‘rescattered’ electrons. I believe only the rescattered electrons experience more significant Coulomb interaction than the case of the circularly polarized field. Can you rule out that you see interference signal only from the direct electrons? Is the contrast in the linear case similar to the contrast in the circular polarization?

Reviewer #3 (Remarks to the Author):

The authors addressed most of my comments in an adequate way. They have extended their manuscript and especially the supplements, which helps in understanding their work.

There is however one point that I think needs further clarification. It concerns answer#1 to reviewer #3 as well answer#6 to reviewer #2. Since the advanced theory shows distinct features that differ significantly from the experiment (supplementary Figure 3 and 4), it is, in my opinion, not valid to state ‘This contrast, as the more complete theory reveals, is mainly due to interaction of the outgoing electron with the neighbouring atom in the dimer (see Supplementary Note 2).’ I think a weaker statement has to be used, and the differences between experiment and theory have to be explicitly addressed in Supplementary note 2, along with their speculation about the origin of the differences (as outlined in answer#6 to reviewer #2).

Replay to reviewers

for the manuscript entitled "**Double-slit photoelectron interference in strong-field ionization of the neon dimer**"

We would like to thank referees for the new comments and suggestions. In the following we address all points of criticism and answer the raised questions. The referee's questions given in blue are followed by our replies in black.

REVIEWERS' COMMENTS:

Reviewer #1 (Remarks to the Author):

I am content that the issues raised in my original report have been addressed and that the manuscript is now publishable in Nature Communications.

Reviewer #2 (Remarks to the Author):

The authors have satisfactorily answered my and the other referees questions. I recommend publication in Nat. Comms. with a few additional clarifications:

1. Comparison to theory (page 6) – “This contrast, as the more complete theory reveals, is mainly due to interaction of the outgoing electron with the neighboring atom in the dimer” : Since the agreement with the ‘more complete theory’ is not very good, a less strong statement is required here.

The mentioned by the referee sentence as well as the corresponding one in the conclusion have been changed: “This contrast, as the more complete theory indicates, is most likely due to interaction of the outgoing electron with the neighbouring atom in the dimer (see Supplementary Note 2).” and “The finite contrast in the interference pattern is most likely related to interaction of the liberated electron with the neighbouring neutral atom in the dimer.”

2. Results of linear polarization:

a.“.. shows that the two-center interference also survives among low energy electrons” – What is considered low energy? Are the electrons measured all / mostly low energy? (Impossible to know from this data, since the third momentum axis is missing)

The experiment with linear polarization provides access to electrons in a wide momentum range from about 2 a.u. down to 0 a.u., which is different for the case of circular polarization, where electron momenta are higher than 1 a.u. The low energy electrons (in strong field ionization usually relates to few eV, or momenta lower than 0.5 a.u.) experience stronger interaction with the Coulomb field of the residual ion. Despite this interaction, even the electrons with momenta around 0 a.u. show (destructive) interference (see the distribution in Fig. 3a). We have specified in parentheses that we mean the electrons with momenta lower than 0.5 a.u. Now the sentence reads: “The experiment with the linearly

polarized light shows that the two-center interference also survives among low energy electrons ($k < 0.5$ a.u.) despite their strong Coulomb interaction with the residual ion."

b. "... despite their strong Coulomb interaction with the residual ion" - In linear polarization there are two contributions to the signal – 'direct' and 'rescattered' electrons. I believe only the rescattered electrons experience more significant Coulomb interaction than the case of the circularly polarized field. Can you rule out that you see interference signal only from the direct electrons? Is the contrast in the linear case similar to the contrast in the circular polarization?

Indeed, the rescattered electrons experience the strongest interaction with an ion (as the rescattering process suggests). The energies of these electrons are however higher than $2U_p$, where $U_p = \frac{e^2}{8\pi^2 \epsilon_0 m_e c^3} \cdot I \lambda^2 \rightarrow U_p [eV] = 9.337 * I \left[10^{14} \frac{W}{cm^2} \right] * (\lambda [\mu m])^2$ is the ponderomotive energy. Given a laser intensity of $I = 1.2 \cdot 10^{15} \frac{W}{cm^2}$ and a central wavelength of $\lambda = 780 nm$, $2U_p = 136 eV$ that corresponds to a momentum of 3.17 a.u. Thus, the distribution in Fig. 3 consists mainly of direct electrons. However, the interaction degree of direct electrons with the parent ion depends on the electron energy (see also the previous answer). The low energy electrons are known to be born at the maximum of the laser field, implying that the tunnel exit is closer to the residual ion than in the case of high energy electrons. Thus, low energy electrons start the quiver motion in the laser field at higher Coulomb potentials. In addition, the low energy electrons stay longer in the vicinity of the ion. These two reasons make interaction of the low energy electrons with the residual ion stronger with respect to the high energy ones. In this context, the experiment with linearly polarized light is different from the circularly polarized one, and provides information about interference of electrons in the wider energy range.

The contrast in case of linear polarization is about 2.0-2.4 (as seen from Fig. 3) and thus similar to that of circular polarization (2.9 and 2.4 for the direct and indirect fragmentation channels, respectively, see supplementary note 6).

Reviewer #3 (Remarks to the Author):

The authors addressed most of my comments in an adequate way. They have extended their manuscript and especially the supplements, which helps in understanding their work.

There is however one point that I think needs further clarification. It concerns answer#1 to reviewer #3 as well answer#6 to reviewer #2. Since the advanced theory shows distinct features that differ significantly from the experiment (supplementary Figure 3 and 4), it is, in my opinion, not valid to state 'This contrast, as the more complete theory reveals, is mainly due to interaction of the outgoing electron with the neighbouring atom in the dimer (see Supplementary Note 2).' I think a weaker statement has to be used, and the differences between experiment and theory have to be explicitly addressed in Supplementary note 2, along with their speculation about the origin of the differences (as outlined in answer#6 to reviewer #2).

We have made the statement weaker as was also suggested by reviewer #2. The discussion from answer #6 to reviewer #2 has been added to Supplementary Note 2.